# Human Brain Endothelial Cell-Derived Extracellular Vesicles Reduce *Toxoplasma gondii* Infection In Vitro in Human Brain and Umbilical Cord Vein Endothelial Cells

**DOI:** 10.3390/ijms26062640

**Published:** 2025-03-14

**Authors:** Luiz Fernando Cardoso Garcia, Victoria Cruz Cavalari, Pryscilla Fanini Wowk, Letusa Albrecht

**Affiliations:** 1Laboratório de Pesquisa em Apicomplexa, Instituto Carlos Chagas, Fundação Oswaldo Cruz (FIOCRUZ-PR), Curitiba 81350-010, Brazil; fgtc@outlook.com (L.F.C.G.); viccavalari97@gmail.com (V.C.C.); 2Grupo de Imunologia Molecular, Celular e Inteligência Artificial, Instituto Carlos Chagas, Fundação Oswaldo Cruz (FIOCRUZ-PR), Curitiba 81350-010, Brazil; pryscilla.wowk@fiocruz.br

**Keywords:** endothelial dysfunction, *Toxoplasma gondii*, extracellular vesicles, host–pathogen interaction, infection modulation

## Abstract

The endothelial layer, formed by endothelial cells, performs crucial functions in maintaining homeostasis. The endothelial integrity and function might be compromised due to various causes, including infection by *Toxoplasma gondii*, leading to an endothelial dysfunction. *Toxoplasma gondii* is an Apicomplexa parasite that infects a broad range of animals, including humans. This parasite can invade all nucleated cells, as well as endothelial cells. The interaction between this protozoan and endothelial cells can be mediated by different molecules, such as extracellular vesicles (EVs), which may either favor or hinder the infectious process. To investigate this interaction, we evaluated the infection of *T. gondii* on human brain microvascular endothelial cells (HBMEC) and human umbilical vein endothelial cells (HUVEC), in addition to assessing transcriptional changes. We also featured the EVs secreted by *T. gondii* and by infected and non-infected HBMEC and HUVEC. Finally, we evaluated the infection of cells stimulated with EVs of parasitic or cellular origin. Our results demonstrated that HUVEC not only exhibit a higher infection rate than HBMEC but also display a more pro-inflammatory transcriptional profile, with increased expression of interleukin-6 (*IL6*), interleukin-8 (*IL8*), and monocyte chemotactic protein-1 (*MCP1*) following infection. Additionally, we observed few differences in the concentration, distribution, and morphology of EVs secreted by both cell types, although their properties in modulating infection varied significantly. When cells were EVs stimulated, EVs from *T. gondii* promoted an increase in the HBMEC infection, EVs from infected or uninfected HBMEC reduced the infection, whereas EVs from HUVEC had no effect on the infectious process. In conclusion, our data indicate that *T. gondii* infection induces distinct changes in different endothelial cell types, and EVs from these cells can contribute to the resolution of the infection.

## 1. Introdution

Endothelial cells form the endothelium, the inner lining of blood and lymphatic vessels, playing vital roles in vascular health and processes such as inflammation, angiogenesis, and the regulation of blood flow [1,2,3]. The endothelial cell barrier is a critical component of the vascular system, acting as a selective interface that regulates the exchange of substances between the bloodstream and surrounding tissues. It is highly selective, allowing the controlled passage of ions, small molecules, and, under specific conditions, larger molecules or cells, such as leukocytes [4,5,6,7]. This selective permeability is crucial for maintaining tissue homeostasis and adapting to physiological demands. In response to inflammatory stimuli, endothelial cells modulate their permeability, enabling immune cells to migrate into tissues during pathogen invasion or tissue damage [8,9,10]. Additionally, the endothelium maintains vascular homeostasis by regulating vascular tone and blood flow through the release of factors that induce vasodilation or vasoconstriction, ensuring blood supply meets metabolic needs [11,12,13,14].

However, the endothelial barrier is susceptible to dysfunction in pathological conditions. During inflammation or infection, inflammatory mediators disrupt junction stability, leading to excessive fluid leakage into tissues [15,16]. Pathogens can directly infect endothelial cells, further exacerbating barrier dysfunction [17,18,19,20,21]. Oxidative stress, often associated with chronic conditions, impairs endothelial cell function by altering junction integrity and increasing permeability, underscoring the vulnerability of the endothelial barrier to various stressors [22,23].

*Toxoplasma gondii*, a protozoan parasite, interacts with endothelial cells during infections, crossing the endothelial barrier to replicate and evade the immune system [24,25,26,27]. Its replicative form, tachyzoites, can directly infect endothelial cells, crossing the endothelial barrier while evading immune detection through mechanisms such as adhesion to infected leukocytes [28,29]. This interaction upregulates cell adhesion molecules like ICAM-1, increasing leukocyte adhesion and inflammation [25,28,29]. Infection induces a pro-inflammatory environment characterized by cytokine release (e.g., IL-6, TNF-α), which recruits immune cells but also contributes to endothelial dysfunction by increasing permeability and causing vascular damage [25,30,31,32,33]. Elevated inflammatory markers and oxidative stress in individuals with acute toxoplasmosis highlight the significant impact of *T. gondii* on endothelial health [21,34].

The interaction between *T. gondii* and host cells has been extensively studied, with extracellular vesicles (EVs) emerging as critical mediators of cellular communication. EVs are lipid bilayer-enclosed particles released by prokaryotic and eukaryotic cells, playing essential roles in intercellular communication and biological processes [35,36,37,38].

EVs contribute to maintaining homeostasis, modulating immune activity, and responding to stressors [39,40]. In the context of *T. gondii* infections, EVs play an important role in pathogen–host interactions. Studies showed that *T. gondii* infection induced the release of neuronal extracellular vesicles, termed TINEVs, which alter neurotransmission, neuroinflammation, and host behavior [41]. These vesicles regulated noradrenaline levels, inhibited dopamine beta-hydroxylase (DBH), and induced gene hypermethylation, highlighting their role in transcriptional regulation [41].

During immune responses to *T. gondii* infection, EVs mediate intercellular communication by transporting molecules such as microRNAs (miRNAs) and proteins. These vesicles stimulate pro-inflammatory responses in host cells, including the production of IL-10, TNF-α, and iNOS in murine macrophages, aiding the host’s immune defense [42,43]. EVs derived from *T. gondii* also contain miRNAs and proteins recognized by the immune system, activating critical anti-infective responses [44,45]. These vesicles transmit infection signals between cells, shaping inflammatory and immune responses during toxoplasmosis.

Beyond modulating host immunity, *T. gondii* itself secretes EVs that actively influence host cell functions. These vesicles carry parasite-derived proteins, lipids, and non-coding RNAs, which interact with host signaling pathways to alter immune responses and cellular processes [46,47]. *T. gondii*-derived EVs have been implicated in suppressing pro-inflammatory cytokine production, modulating antigen presentation, and interfering with apoptosis, ultimately facilitating parasite survival [42,48]. Additionally, they may contribute to parasite invasion and persistence by promoting cytoskeletal rearrangements and metabolic shifts that create a more favorable intracellular environment [32,42].

The secretion of EVs by *T. gondii* may be linked to the parasite’s broader secretory mechanisms. Gendrin et al. (2008) investigated how *T. gondii* employs unconventional pathways to transport transmembrane proteins into the host vacuole, revealing that some secreted proteins bypass the conventional endoplasmic reticulum and Golgi apparatus routes, suggesting specialized export mechanisms [48]. While their study does not directly describe EV release, these findings provide insights into the parasite’s secretory processes and its ability to modulate the host’s intracellular environment. Understanding these mechanisms may contribute to future investigations into EV biogenesis in *T. gondii*, helping to elucidate their role in host–parasite interactions.

The ability of *T. gondii* to secrete EVs highlights their potential role in immune evasion and the establishment of chronic infection, further emphasizing the complexity of host–pathogen interactions. Here, we focus on EVs-mediated host–pathogen interactions and evaluate the role of EVs derived from *T. gondii*-infected and non-infected endothelial cells on *T. gondii* invasion.

## 2. Results

### 2.1. HBMEC and HUVEC Cells Respond Distinctly to Toxoplasma gondii Infection

Endothelial cells are considered a replicative niche for *T. gondii* [25]. To investigate whether different endothelial cell types exhibit similar infection profile, we analyzed *T. gondii* infection, evaluating the number of intracellular tachyzoites in HBMEC and HUVEC by Fluorescence Microscopy (FM) (Figure 1A) and Flow Cytometry (FC) (Figure 1B,C). Our findings revealed that the infection rate in HUVEC is approximately 1.52 times higher than in HBMEC (*p*-value = 0.0096) (Figure 1D). An average, HUVEC had 4.57 tachyzoites per cell, while the infected HBMEC had 1.52 tachyzoites per infected cell (*p*-value = 0.0092) (Figure 1E). Additionally, over 60% of both infected HBMEC and HUVEC contained a single intracellular tachyzoite (Figure 1F).

To investigate whether these differences were influenced by the culture medium, we cultured HBMEC in EBM™-2, a rich medium typically used for HUVEC. Surprisingly, this medium reduced the infection rate in HBMEC, contrary to the increase observed in HUVEC (*p-*value *=* 0.0048) (Figure 1G). However, the number of intracellular tachyzoites per cell remained unchanged (Figure 1H,I). These findings were confirmed using FC, which again showed a higher infection rate in HUVEC compared to HBMEC (*p-*value *=* 0.0017) (Figure 1J).

These results emphasize the role of intrinsic cellular properties in modulating susceptibility to *T. gondii* infection. While the culture medium impacts some parameters, our data suggest that infection rates are primarily governed by cell-specific characteristics, paving the way for further studies on the cellular and molecular mechanisms underlying these differences.

### 2.2. HBMEC and HUVEC Cells Have Characteristic Gene and Protein Changes in T. gondii Infection

In order to better understand the interaction between *T. gondii* and infected endothelial cells, markers of endothelial activation and dysfunction were evaluated. When evaluating transcriptional changes individually, no statistically significant changes were identified in HBMEC for almost all of the genes evaluated throughout the infection process (Figure 2A–I). Notable exceptions included *ADAMTS13* (Figure 2A), *ICAM1* (Figure 2C), and *MCP1* (Figure 2F), which showed reduced expression 2 h (*ADAMTS13* and *ICAM1*) and 4 h (*MCP1*) post-infection, to below detectable levels by qPCR.

In HUVEC, transcriptional increases in *IL6*, *IL8*, *MCP1*, *SOCS3*, and *VEGF* were observed. *IL6* and *IL8* showed significant changes at 4 h post-infection (*p*-value = 0.0417 and *p*-value = 0.0115, respectively) (Figure 2M,N), while *MCP1* and *SOCS3* exhibited changes from 1 and 2 h post-infection, respectively (*p*-value < 0.05). Although *SOCS3* levels increased at 2 h post-infection (*p*-value = 0.0097), they returned to baseline levels by 4 h post-infection (Figure 2Q). *MCP1* showed the most significant change, with a 25-fold increase in expression compared to basal levels (*p*-value = 0.0007) (Figure 2O). For *VEGF*, no significant differences were observed when compared to uninfected cells. However, a slight reduction at 1 h was followed by increases at 2 h (*p*-value = 0.0417) and 4 h (*p*-value = 0.0460) post-infection (Figure 2R).

*ADAMTS13* transcripts were detectable in both HBMEC and HUVEC (Figure 2W), but no significant changes were observed. Transcript levels became undetectable 2 h post-infection in HBMEC and 4 h post-infection in HUVEC. Finally, genes including *IL1A*, *IL1B*, *IL10RA*, *IL10RB*, *SELECTINE*, *TNF* were undetectable in infected and uninfected HBMEC and HUVEC. These findings reinforce the differences between HBMEC and HUVEC in their responses to *T. gondii* infection. While HUVEC displayed a pronounced pro-inflammatory profile, no significant transcriptional changes were identified in HBMEC.

We then divided the genes into five categories: inflammation, dysfunction, coagulation, angiogenesis, and adhesion. It was observed an increasing transcriptional change in HUVEC after infection with *T. gondii* across all categories, while in HBMEC, a negative regulation of adhesion-related genes was observed (Figure 2S–V). Gene changes in HBMEC and HUVEC throughout the infection process could be observed by a heatmap in which the upregulation of different genes could be seen occurring more frequently in HUVEC than in HBMEC (Figure 2W). The difference in gene expression can also be observed in more detail in table format (Appendix A).

To further explore these differences, ICAM-1 expression was evaluated at the protein level in both HBMEC and HUVEC using flow cytometry. Figure 3A shows the analysis of infected and uninfected cells expressing ICAM-1. The total number of ICAM-1-positive cells increased significantly in the presence of *T. gondii*. HBMEC ICAM-1 expression increased from ~62.23% to ~74.20% (*p*-value = 0.0173) (Figure 3B), while HUVEC expression rose from ~7.67% to ~18.49% (*p*-value = 0.0011) (Figure 3C).

Mean fluorescence intensity (MFI) analyses showed increased ICAM-1 expression per cell in both HBMEC and HUVEC after contact with *T. gondii* (1.433 to 2.100 and 0.300 to 0.567, respectively; *p*-value < 0.0001 and *p*-value = 0.0013) (Figure 3D–G).

Although ICAM-1-positive cells increased, uninfected cells that contacted the parasite did not exhibit notable changes (~62.7% of HBMEC and ~6.4% of HUVEC) (Figure 3A). Despite transcriptional changes observed within 4 h post-infection, ICAM-1 protein levels were measured 24 h post-infection. Contrary to expectations, gene expression of *ICAM1* decreased to undetectable levels 2 h post-infection in HBMEC (Figure 3H) and 8 h post-infection in HUVEC (Figure 3I). We also evaluate the expression of *VCAM1* and we observed that in HUVEC it increases, reaching a peak at hour 4 post-infection (7.5-fold comparing to uninfected cells) (*p*-value < 0.0001) and begins to reduce from hour 8 post-infection (Appendix A). We were unable to identify detectable levels *VCAM1* at HBMEC. Upon discovering that infected cells exhibit higher expression of surface ICAM-1, we speculated that this protein might facilitate the attachment of more parasites to the cells, as observed in retinal endothelium [49]. Since TNF-α can stimulate ICAM-1 expression, we evaluated the infection rate of HBMEC and HUVEC in the presence of TNF-α. However, no significant difference was observed (Appendix A).

These results reveal notable differences in how HBMEC and HUVEC respond to *T. gondii* infection. While HUVEC showed a strong pro-inflammatory response, HBMEC displayed more subdued changes, including reduced gene expression. Both cell types exhibited increased ICAM-1 expression, even without direct infection, indicating that contact with *T. gondii* triggers cellular responses. These findings highlight the complexity of *T. gondii*-endothelial interactions and call for further study of the underlying mechanisms.

### 2.3. HBMEC HUVEC and T. gondii Secret EVs with Similar Size and Distribution

Seeking to better understand the mechanisms of cellular communication between endothelial cells and *T. gondii*, we evaluated the EVs shedded from the parasite and from infected or uninfected endothelial cells. We observed that HUVEC secrete EVs with an average diameter between 100 and 200 nanometers, and after *T. gondii* infection, the particle distribution changes to an average between 100 and 300 nanometers (Figure 4A–C). On the other hand, HBMEC secrete EVs with an average diameter between 100 and 300 nanometers, whether infected or not (Figure 4D–F). EVs secreted by uninfected HUVEC (HU-EV) or HBMEC (HB-EV), 2 and 24 h post-infection (2 h.p.i. and 24 h.p.i., respectively) did not present morphological differences in the particles throughout the infectious process (Figure 4G–L).

After this preliminary evaluation of EVs originating from endothelial cells, we repeated the analysis with EVs from *T. gondii (Tg-EV)*, and we observed, using TEM, particles with a concave (cup-shaped) morphology indicative of EVs (Figure 4M). Using NTA, we verified a less diffuse distribution compared to EVs secreted by endothelial cells and with an average particle diameter of approximately 200 nanometers (Figure 4N). Finally, we compared the concentration of EVs secreted by HUVEC and HBMEC throughout the infectious process (HUTg-EV and HBTg-EV respectively), including Tg-EV. HUVEC secrete ~3.0 × 10^7^ particles/mL (Figure 4O), HBMEC ~2.6 × 10^7^ particles/mL (Figure 4P), and *T. gondii* ~1.2 × 10^7^ particles/mL (Figure 4Q). There was no significant difference between all comparisons.

Using mass spectrometry, we identified classic EV markers in particles secreted by HUVEC (Annexin A2, Heat Shock Protein 70 or HSP70, and Cluster of Differentiation 44 or CD44), HBMEC (Annexin A2 and CD44), and *T. gondii* (HSP70, Annexin A2, and CD44) (Appendix A).

Aiming to better understand Tg-EV, we evaluated tachyzoites by TEM in search of events suggestive to the secretion or uptake of EVs. We observed structures indicating EVs close to the apical complex of tachyzoites (Figure 5A), events indicating an exchange of EVs (Figure 5B), or even accumulation between different parasites (Figure 5C). We also evaluated the tachyzoites using Scanning Electron Microscopy (SEM), looking for structures on the surface of the parasites with EVs morphology, and we were able to identify particles of different sizes along the entire surface of the parasites (Figure 5D–F).

### 2.4. T. gondii EVs Can Affect Gene Transcription and Endothelial Cell Viability

Once we verified that tachyzoites secrete EVs and we managed to isolate and characterize them properly, our next step was to evaluate the effects of these particles on endothelial cells. From this point on, as we found a less pro-inflammatory profile in HBMEC, we focused experiments on these cells. We started by checking the viability of HBMEC with different concentrations of Tg-EV, and we observed that the exposure of cells to EVs concentrations starting at 40 μg/mL begins to have a cytotoxic effect on target cells (*p*-value = 0.0279) (Figure 6A). Based on this, we evaluated *T. gondii* infection rate when HBMEC were pre-incubated with 0.001 μg/mL (approximate value secreted by 3.2 × 10^4^ tachyzoites) and 20.0 μg/mL (highest non-cytotoxic concentration), and did not observe significant differences (*p*-value = 0.3118) (Figure 6B).

Next, we checked the expression of *IL8*, *MCP1*, *ENOS*, and *VCAM1* in HBMEC and HUVEC exposed to 0.001 μg/mL EVs. We found that HBMEC incubated for 24 h showed a negative regulation of *IL8* (2.17-fold) (*p*-value = 0.0060) (Figure 6C), a positive regulation of *MCP1* (2.156-fold, *p*-value = 0.0165) (Figure 6D), and *ENOS* (1.2-fold, *p*-value = 0.0010) (Figure 6E). For HUVEC, we verified a negative regulation of *IL8* (111.10-fold, *p*-value < 0.0001) (Figure 6F), *MCP1* (4.4-fold, *p*-value < 0.0001) (Figure 6G), and *VCAM1* (6.54-fold, *p*-value < 0.0001) (Figure 6H). We also evaluated the expression of *IL6* in HBMEC but did not detect significant differences. *VCAM1* in HBMEC or *ENOS* in HUVEC transcripts were under detectable levels.

The significantly altered inflammatory marker expression, particularly in HBMEC, with *IL8* downregulation and *MCP1* and *ENOS* upregulation due to exposure to Tg-EV suggests that Tg-EVs play a role in modulating inflammation and endothelial function during infection.

### 2.5. HBMECs Supernatant Contains EVs That Promote a Protective Effect in the Infectious Context on HBMEC and HUVEC

To better understand the differences in infection rates between HBMEC and HUVEC, we infected HBMEC with *T. gondii* after incubating the cells with Tg-EV (Figure 7J–L), HB-EV (Figure 7A–C), HBTg-EV (Figure 7D–F), and the secretome from infected HBMEC (HBTg-SEC) (Figure 7G–I). We observed that pre-stimulating of HBMEC with both HBTg-EV and HBTg-SEC led to a reduction in the infection rate (from ~12.33% without EVs pre-treatment to ~6.91% after HBTg-EV or ~7.14% with HBTg-SEC treatment, *p*-value = 0.0047 and 0.0146, respectively) and intracellular tachyzoites (from ~2.720 without EVs pre-treatment to ~1.215 after HBTg-EV and ~1.239 with HBTg-SEC treatment, *p*-value = 0.0002 and 0.0003, respectively) (Figure 7D–F and 7G–I).

Interestingly, when stimulating cells with HB-EV, we also observed an influence on infection dynamics. Although this treatment did not reduce the number of infected cells (*p*-value = 0.0942) (Figure 7A), it reduced the number of intracellular tachyzoites (from ~2.720 without EVs pre-treatment to ~1.280, *p*-value = 0.0003) (Figure 7B,C). Finally, when stimulating HBMEC with Tg-EV, we observed an increase in the infection rate (from ~12.33% without EVs pre-treatment to ~20.53%, *p-value <* 0.0001), accompanied by a reduction in the number of intracellular tachyzoites (from ~2.720 without EVs pre-treatment to ~1.842, *p*-value = 0.0017).

Once a modulatory effect conferred by HBMEC EVs was suggested by these results, we investigated whether this could be applied to other cell types such as HUVEC. Similar to HBMEC, when we stimulated HUVEC with HBTg-EV and HBTg-SEC, we observed a reduction in the infection rate (from ~22.91% without EVs pre-treatment to ~3.885% with HBTg-EV and ~4.345% with HBTg-SEC treatment, *p*-value = 0.0031 and 0.0038, respectively) (Figure 8D,G). However, there was no reduction in the number of intracellular tachyzoites (Figure 8E,F,H,I). A reduction in the infection rate was also observed when cells were stimulated by HB-EV (from ~22.91% without EVs pre-treatment to ~4.992%, *p*-value = 0.0048) (Figure 8A). Although no significant differences were found in the average number of intracellular tachyzoites (*p*-value = 0.0549), more cells with only a single tachyzoite were identified (from ~47.01% without EVs pre-treatment to ~79.63%, *p <* 0.0001) (Figure 8G). No changes were observed in the infection rate of HUVEC stimulated with Tg-EV (Figure 8J). The only event noted was a reduction in the number of cells with two tachyzoites (from ~34.88% without EVs pre-treatment to ~10.22%, *p*-value = 0.0071) (Figure 8L).

Finally, we conducted a comparative analysis of all EV treatments in both HBMEC and HUVEC against the PBS group (vehicle) (Figure 9A–D). We observed similar results between HBTg-SEC and HBTg-EV, with reductions in the number of infected cells and intracellular tachyzoites in HBMEC (Figure 9A and 9B, respectively), and HUVEC (from ~22.91% to ~4.11%) (Figure 9C and 9D, respectively) compared to cells incubated with the vehicle (PBS). As seen before, incubation of HBMEC with HB-EV did not reduce the rate of infected cells (*p*-value = 0.0942) (Figure 9A), but affected the number of intracellular tachyzoites (*p*-value = 0.0003) (Figure 9B). Otherwise, in HUVEC, HB-EV reduced both the infection rate and number of intracellular tachyzoites (Figure 9C,D). Incubation of HBMEC with Tg-EV increased the infection rate (~20.40%) compared to all treatments (Figure 9A), while no similar effect was observed in HUVEC, which maintained high levels of infection (~23.22%) as when incubated with PBS (~22.91%) (Figure 9C). Interestingly, Tg-EV had distinct effects on intracellular tachyzoite counts when comparing HBMEC with HUVEC, while tachyzoite counts were reduced in HBMEC compared to PBS (from ~2.72% to ~1.86%) (Figure 9B); no effect was observed in HUVEC (Figure 9D).

After confirming the effect of HB-EV and HBTg-EV in HBMEC and HUVEC infection dynamics, we investigated whether HUVEC could also secrete EVs with a similar or different effect. HU-EV, HUTg-EV, or Tg-EV did not alter the infection rate in HUVEC (Figure 9E). Similarly, no significant differences were identified in the mean (Figure 9F) or absolute (Figure 9G) counts of intracellular tachyzoites in HUVEC incubated with HUTg-EV.

Finally, given that HB-EV promoted a similar effect in both HBMEC and HUVEC infection dynamics, we investigated whether varying its concentration would affect the infection rate (Figure 9H) or intracellular tachyzoites (Figure 9I,J) in HUVEC. We found that incubation with 0.01, 0.1, and 1.0 μg/mL of HB-EV caused similar reductions in infection rates comparing to the control (*p*-value < 0.05) (Figure 9H). Interesting, only 1.0 μg/mL HB-EV reduced the average number of intracellular tachyzoites (Figure 9I).

These results highlight the potential of EVs derived from HBMEC to reduce *T. gondii* infection, emphasizing important differences in responses between HBMEC and HUVEC. While HBMEC displayed a reduction in intracellular tachyzoite loads (from ~2.720 without EVs pre-treatment to ~1.280), HUVEC show a more variable response, with a more robust reduction in infection rate (from ~22.91% without EVs pre-treatment to ~4.992%) and tachyzoite distributions, but with no difference in tachyzoite load. Additionally, the lack of effect from HUVEC EVs underscores the functional specificity of HBMEC EVs. These findings suggest that HBMEC-derived EVs, particularly HB-EV and HBTg-EV, present immunomodulatory properties that could be leveraged for therapeutic strategies against toxoplasmosis.

## 3. Discussion

Endothelial cells interface between blood and tissues, making them key targets for infectious agents, including *T. gondii* [21,50]. This protozoan can invade and replicate in endothelial cells, with susceptibility varying by anatomical region [51]. While different studies focus on its ability to cross the endothelial barrier, broader endothelial responses to infection remain underexplored [24,52].

Konradt et al., 2016, observed that brain endothelial cells exhibit lower *T. gondii* infection rates than lung endothelial cells [24]. Similarly, HUVEC display lower infection rates than HMEC (human microvasculature endothelial cells). Interestingly, the infection rate depends not only on the cell type but also on the *T. gondii* strain. For example, the less virulent ME-49 strain resulted in higher HUVEC infection rates than the more virulent RH strain [53]. These findings suggest that endothelial cell susceptibility to *T. gondii* infection may depend on cell-specific properties and parasite strain characteristics.

Further supporting this notion, researchers hypothesized that the rate of *T. gondii* replication might correlate with the host cell cycle. The parasite preferentially invades cells in the S phase of the cell cycle, leading to higher infection rates in rapidly dividing cells [54]. Although our study did not directly compare the cell cycle of HBMEC and HUVEC, it was noted that individual HUVEC sometimes harbored up to eight tachyzoites, while neighboring cells remained uninfected. This observation aligns with previous reports suggesting that infected cells might secrete factors that inhibit the cell cycle of nearby cells [55].

Despite both being endothelial cell types, HBMEC and HUVEC respond differently to *T. gondii* infection, not only in terms of infection rates but also in transcriptional changes related to cytokine production, adhesion molecules, and angiogenesis [25]. Importantly, differences observed between HBMEC and HUVEC in our study were not attributed to variations in culture media, possibly due to intrinsic cellular differences.

According to our results, HUVEC displayed a higher inflammatory response than HBMEC, evidenced by significantly elevated levels of *IL6*, *IL8*, *SOCS3*, and *MCP1*. This observation aligns with the literature indicating that HUVEC mount robust inflammatory responses to different pro-inflammatory stimuli [25,52,56].

MCP1 is a chemoattractant molecule primarily involved in monocyte recruitment but also implicated in inflammation and apoptosis [57,58]. In our study, *MCP1* expression increased continuously over time in HUVEC infected with *T. gondii*, suggesting sustained stimulation by the parasite. Similar results were reported in retinal endothelial cells infected with *T. gondii* tachyzoites, where *MCP1* expression remained elevated 24 h.p.i. [21]. Interestingly, *MCP1* expression patterns differ in fibroblasts exposed to *T. gondii*. When fibroblasts were incubated with purified parasitic proteins such as SAG1, *MCP1* expression was intense but transient. In contrast, during active *T. gondii* infection, *MCP1* expression persisted for 24 h [59]. These findings underscore the parasite’s ability to induce prolonged *MCP1* expression across various cell types, including HUVEC. Conversely, HBMEC showed markedly different *MCP1* expression dynamics. While *MCP1* levels remained constitutive during the first 2 h.p.i., they became undetectable by 4 h.

HBMEC *MCP1* expression varies depending on the infectious agent: it increases during Hepatitis C infection [60], remains basal in Epstein–Barr virus infection [61], and shows distinct patterns in *Cryptococcus neoformans* infection [62]. MCP1 plays a key role in inflammation and pathogen elimination. Dysregulated MCP1 production is linked to neuroinflammation [63], multiple sclerosis [64], and can facilitate HIV1 migration to the fetal brain [65] or trigger spontaneous abortion [66].

We also examined *TNF* expression, noting that IFN-γ and TNF-α stimulate endothelial cells to combat pathogens like *Pseudomonas aeruginosa* via ROS production [17]. However, ROS has minimal impact on limiting *T. gondii* proliferation in HUVEC [67]. TNF expression in HUVEC is triggered by LPS, but secretion remains intracellular [68]. Our experiments showed no detectable *TNF* in HUVEC or HBMEC during *T. gondii* infection, consistent with other studies [69]. *ENOS* expression, influenced by IFN-γ and TNF-α, may affect *T. gondii* defenses [34], but excessive nitric oxide can cause endothelial dysfunction [70]. Our findings match studies showing suppressed *ENOS* expression in HUVEC to maintain pregnancy [71], with no modulation observed in HBMEC.

ADAMTS13, primarily involved in coagulation, has neuroprotective roles [72] and its expression in endothelial cells limits *T. gondii* migration to the CNS. However, post-infection downregulation was observed, likely due to tachyzoites or altered *VWF* expression [73], though undetectable *VWF* levels prevented further analysis. For HUVEC, maintaining basal *ANG1* and *ANG2* expression is crucial for neovasculogenesis [74]. Our results support this stability, with slight *VEGF* upregulation noted, which could disrupt neovasculogenesis. This contrasts with a study linking reduced *VEGF* in trophoblasts to pregnancy loss in *T. gondii* seropositive women [75], highlighting cell-specific responses [76].

The expression of adhesion molecules, such as ICAM-1 and VCAM-1, facilitates leukocyte recruitment to inflammation sites [50]. Modulation of these molecules involves various factors, including VEGF [77], microRNAs associated with lipoproteins [78], or hydrocortisone conjugated to glycine [79]. Infection with *T. gondii* enhances ICAM-1 expression on endothelial surfaces [25], as confirmed in our study. Yet, transcriptional levels of *ICAM1* mRNA diminished post-infection in HUVEC by hour 8 and in HBMEC by hour 2. This protein-mRNA discrepancy suggests a role for ICAM-1 recycling from lysosomes, given its involvement in endocytosis and leukocyte–endothelium interactions [80]. ICAM-1 expression is linked to *T. gondii* migration through retinal endothelial cells [49], raising the possibility of its involvement in parasite entry [53]. However, our results showed no increase in endothelial infection rates despite heightened ICAM-1/VCAM-1 levels, suggesting that these molecules are not used by *T. gondii* for invasion. Additionally, TNF-α activation, while promoting ICAM-1/VCAM-1 expression and tachyzoite clearance pathways, did not significantly influence intracellular tachyzoite counts at 1 h.p.i. This aligns with reports that eNOS’s acute protective role is minimal [31] and IDO is activated by IFN-γ rather than TNF-α [81].

Lachenmaier and her team also observed an increase in the expression of ICAM-1, IL6, and MCP1 in brain endothelial cells infected with T. gondii [25]. Other studies have demonstrated changes in MCP-1 and RANTES (Regulated upon Activation, Normal T Cell Expressed and Secreted) in rat retinal endothelial cells [21]. Additionally, changes related to the deregulation of barrier function, such as cytoskeleton reorganization through Hippo signaling in human endothelial cells infected with T. gondii, have also been described [32]. Although these studies utilize different types of endothelial cells or even different model species, the similarities in the responses observed highlight a significant pro-inflammatory response in endothelial cells infected with T. gondii.

For years, the endothelium was considered a uniform cell group [82], but recent advances reveal that endothelial cells from different regions have distinct activities [56,83,84,85]. Similarly, EVs show variability, with HBMEC and HUVEC producing vesicles of specific sizes and concentrations that change minimally during infection. However, HUVEC EVs showed slight variability after *T. gondii* infection. We hypothesize that the lower infection rate in HBMECs compared to HUVECs may be due to differential cellular communication, especially via soluble factors like proteins or EVs, with HBMEC EVs potentially playing a larger role in immune processes.

Before starting infection experiments on endothelial cells exposed to EVs from different origins, we evaluated whether EVs from *T. gondii* could be cytotoxic to endothelial cells. We found that cell death processes began at concentrations above 40 μg/mL. Interestingly, we observed that HBMECs were resistant even to high concentrations of *T. gondii*-derived EVs, nearly reaching concentrations observed in specialized immune cells like macrophages stimulated with the same type of EVs [43]. With this finding, and knowing that HBMEC and HUVEC *T. gondii* infections induced gene transcription changes, we examined whether *T. gondii*-derived EVs could perform the same effect. Remarkably, we observed a more pro-inflammatory profile in HBMECs and an anti-inflammatory profile in HUVECs stimulated with *T. gondii*-derived EVs.

EVs from infected HBMECs, and even from uninfected ones, appeared to have a modulatory role against infection in both HBMEC and HUVEC cells. The mechanisms behind this effect are still under investigation, but one hypothesis is that specific proteins in EVs derived from HBMECs, such as integrins or collagen, could be remodeling the cell membrane and making the cell less permissive to infection. However, more studies are needed to support this.

Several hypotheses about the protective effects mediated by EVs can be drawn from the current literature. For example, it has been shown that interferons play a crucial role in limiting *T. gondii* replication [86], and evidence suggests that this response can be modulated by EVs [87]. Additionally, EVs released by infected cells, such as in the case of *Mycobacterium tuberculosis*, contain microRNAs that can repress genes associated with inflammation, a mechanism that might can also be applied to *T. gondii* infection [88]. Moreover, EVs from different sources, such as TNF-α-stimulated endothelial cells [89], can induce the release of pro-inflammatory cytokines, such as TNF-α and IL-6, which could be crucial in limiting *T. gondii* infection [90,91].

The concentration of EVs is as important as their source. The 1.17 ng/mL concentration used in our study approximates physiological levels secreted by endothelial cells and was sufficient to elicit an antiparasitic response. This contrasts with studies using higher EV concentrations (μg range) to demonstrate effects like monocyte migration [89] or uptake by various cells [92,93]. Our results suggest that even lower concentrations can trigger biologically significant responses. Notably, EVs from HBMECs modulated both HBMEC and HUVEC, while those from HUVECs did not, supporting the idea that HUVEC EVs are more involved in cell growth and migration. Recent studies on tumor-derived EVs show they can modulate gene expression in endothelial cells, promoting *VEGF*, *VWF*, *VEGFR2*, *CXCL5*, and *CCR1* expression and enhancing endothelial cell proliferation and migration [43,94,95].

While EVs from infected host cells can modulate the infection dynamics in favor of the host, the incubation of cells with EVs of parasitic origin appears to have the opposite effect, as demonstrated in our study. These findings are consistent with previous research. For instance, it has been shown that dendritic cells infected with *T. gondii* produce exosomes that enhance the Th1 immune response [94]. Similarly, exosomes derived from *T. gondii*-infected murine L6 cells contain miRNAs associated with cell cycle arrest [55]. Additionally, macrophages exposed to parasitic exosomes produce higher levels of IL-12, TNF-α, and IFN-γ, and mice exposed to these exosomes exhibit increased survival when challenged with lethal doses of tachyzoites [96]. Chiocolla’s team further investigated *T. gondii* EVs, identifying distinct characteristics among EVs from different strains [47], with potential applications in vaccination strategies [42]. These findings highlight the importance of studying extracellular vesicles from *T. gondii* and their interactions with various cell types, as they may offer valuable insights and opportunities for therapeutic interventions.

Among the promising findings, Pereira-Chiocolla and colleagues demonstrated the protective potential of immunizing mice with *T. gondii*-derived EVs, resulting in increased specific antibodies, elevated IFN-γ, IL-10, TNF-α, and IL-17 expression, reduced parasitemia, and higher survival rates [45]. However, a subsequent study by Chiocolla’s team revealed distinct characteristics of EVs from different *T. gondii* strains, showing that both virulent (RH) and less virulent (ME-49) strains increased parasitemia in RH-infected mice [47]. This contrast likely results from differences in EV stimulation timing—immunization before infection versus simultaneous infection and EV exposure. Our findings align with the second study, as our 24-h EV pre-incubation contrasts with the four-week immunization protocol. These differences may explain endothelial cells’ susceptibility to *T. gondii* infection following parasitic EV stimulation.

At last, in our previous studies, we characterized EVs obtained from HBMEC and identified several proteins that could influence these dynamics. Proteins such as Annexin A1, CD9, ITGB1, and LAMP1 were associated with modulating the infectious process, while others like HSPA1B, HSPA5, CD44, FN1, VIM, and TLN1 were linked to protecting cells against oxidative stress and assisting in cell membrane repair [97]. Further studies to verify the transfer of these proteins via EVs and their specific roles in infection modulation would strongly contribute to understanding the molecular interactions involved in *T. gondii* infection. Figure 10 presents a graphical summary that integratively illustrates the infectious microenvironment, including the interactions between HBMEC, EVs, and *T. gondii*.

In summary, studies on endothelial cell interaction with *T. gondii* reveal intriguing complexities that extend beyond the parasite’s invasiveness. The variation in infection rates between different types of endothelial cells, as well as the potential influence of extracellular vesicles (EVs), raises important questions about the mechanisms that govern cellular responses to infection. Notably, parasitic EVs appear to increase infection rates, while EVs from host cells—whether related to infection or not—exhibit protective effects, potentially reducing infection rates. This dichotomy underscores the importance of understanding the molecular cargo within these vesicles and their specific roles in modulating the infectious process.

## 4. Materials and Methods

The experimental flowchart summarizing the experiments described below is shown in Figure 11.

### 4.1. Cells and Parasite Culture

Immortalized human dermal fibroblast (NHDF) cells, human cerebral microvascular endothelial cells (HBMEC), and human umbilical vein endothelial cells (HUVEC) were used in this study.

Immortalized NHDF (Lonza Group, Switzerland, cat. no. CC2509) and HBMEC (Creative Biolabs, Inc., United States, cat. no. DBFF-1123-HX2043) were maintained in 75 cm^2^ culture flasks (T75) with 10 mL of high glucose DMEM (Dulbecco’s Modified Eagle Medium) (Gibco™, Waltman, MA, USA) supplemented with 10% fetal bovine serum (FBS) (Gibco™, London, UK), 2 mM L-glutamine (Gibco™, UK), 50 U/mL penicillin (Gibco™, UK), and 50 µg/mL streptomycin (Gibco™, K), at 37 °C with 5% CO_2_. Once cells reached 90% confluence, approximately 1.4 × 10^6^ cells per flask were detached using 2.5 mL of trypsin (5 mg/mL) (Sigma-Aldrich™, USA) + EDTA (ethylenediaminetetraacetic acid) (2 mg/mL) (Sigma-Aldrich™, St. Louis, MI, US) and transferred to a new T75 flask at a final concentration of 3.0 × 10^5^ cells. Confluence was typically achieved after seven days of cultivation, with medium changes every two days. Cells were used for a maximum of six passages post-thaw [46].

Primary human umbilical vein endothelial cells (HUVEC) (Cat. No. C2519A) were cultured similarly. They were maintained in T75 flasks with 10 mL of EBM™-2 (Endothelial Cell Basal Medium) (Lonza, Basel, CH), supplemented with 2.5% FBS, 0.02% hydrocortisone, 0.2% HBGF (Human Basic Fibroblast Growth Factor), 0.05% VEGF (Vascular Endothelial Growth Factor), 0.05% R3IGF (Insulin-like Growth Factor-1 LongArginine), 0.05% ascorbic acid, 0.05% HEGF (Human Epidermal Growth Factor), and 0.05% GA-1000 (Gentamicin Sulfate-Amphotericin) (Lonza, Basel, Switzerland), at 37 °C with 5% CO_2_ [98]. Cell replication followed the same procedure as for HBMEC.

Tachyzoites of *Toxoplasma gondii* (ME-49 strain), kindly provided by Dr. Érica dos Santos Martins Duarte (Federal University of Minas Gerais, Brazil), were maintained in six-well plates with NHDF cells at 90% confluence in DMEM high glucose medium with 10% FBS, 2 mM L-glutamine, 50 U/mL penicillin, and 50 µg/mL streptomycin at 37 °C and 5% CO_2_, with medium changes every two days. When 7.0 × 10⁶ parasites were inoculated into 1.4 × 10⁶ NHDF cells (MOI-5), cell lysis occurred around five days later, releasing tachyzoites into the supernatant. These were then maintained in new NHDF cultures or frozen for experiments. To ensure normal metabolic activity, recently thawed parasites were cultured in NHDF cells at least once before use [53].

### 4.2. Fetal Bovine Serum Depletion and Isolation of Extracellular Vesicles

Extracellular vesicles (EVs) were removed from FBS by ultracentrifugation in a Himac CP90WX ultracentrifuge (Hitachi, Tokyo, Japan) at 100,000× *g*, 4 °C, for 18 h. The supernatant was then filtered through a 0.22 µm membrane and stored at −20 °C [99].

Tachyzoites of *T. gondii* were maintained in NHDF cells as described. Upon cell lysis, free tachyzoites were collected and washed with 10 mL of PBS, followed by centrifugation at 1,000 × *g* for five minutes, repeated three times to remove debris. Then, 1.0 × 10^9^ tachyzoites were cultured in T75 flasks with 10 mL of DMEM at 37 °C and 5% CO_2_ for two hours. The supernatant was filtered through a 0.45 µm membrane, and EVs were purified by differential ultracentrifugation in ascending order: 300× *g* for 10 min, 2000× *g* for 10 min, 10,000× *g* for 30 min, and 100,000× *g* for 70 min twice. The precipitate from the final centrifugation cycle was resuspended in 50 µL of sterile PBS and stored at 4 °C for up to seven days [43,46].

EVs from uninfected HBMEC (HB-EV) and HUVEC (HU-EV) were isolated following the same protocol. For infected cells, HBMEC (HBTg-EV) and HUVEC (HUTg-EV) were cultured, and, upon reaching 90% confluence, 1.4 × 10^7^ tachyzoites (MOI-5) were added. After one hour, the supernatant was removed, the cells were washed three times with sterile PBS, and fresh culture medium was added. Cells were incubated for either 2 or 24 h, and EVs were isolated as described [46,100]. The secretome of infected cells (HBTg-SEC) was collected after the first 100,000× *g* centrifugation.

### 4.3. Characterization of Extracellular Vesicles

EVs obtained from the described protocol were diluted 1:10 to obtain a final volume of 500 μL. The EV suspension was then collected using a sterile 1 mL syringe attached to an infusion pump that injected the material at a rate of 50 (~5.2 μL/min) into a NanoSight LM14 system (Malvern, UK) equipped with a 532 nm laser module. Image capture was configured for three 60-s recordings. Between sample exchanges, the infusion system was flushed with ultrapure water until a maximum of one particle per frame was observed by the 3.0 SCMOS system. The results were exported in spreadsheet format and subjected to statistical analysis. At least three replicates were performed for each assessment [46,96].

Another aliquot of the collected EVs was prepared for identification by Transmission Electron Microscopy (TEM). Glass slides were dipped in 0.5% Formvar + chloroform for 10 s to form a thin membrane. The slides were then carefully immersed in a beaker with 20 mL of water to detach the membrane, which remained floating on the surface. Metal grids were positioned on the membrane while still in the water. The set, consisting of the membrane and grid, was removed with the help of a plastic paraffin film cutout and left to dry in a Petri dish for 10 min. Afterward, the neutralization of static electricity was performed using a Zerostat pistol-type anti-static ionizer (Milty, England). Subsequently, 50 μL of PBS containing the extracellular vesicles were applied to the grids, which were left to dry for 1 h in a sealed Petri dish. At the end of this period, excess PBS was removed using a paper towel placed carefully next to the grid. Once dry, negative staining of the samples commenced, starting with another round of static neutralization. After that, 1 mL of uranyl acetate was filtered through a 0.22 μm membrane and carefully applied to a paper filter until a small drop with a circumference similar to that of the metal grids formed. The grids were then carefully placed over the drops of uranyl acetate and left for 20 min, protected from light. Afterward, the grids were carefully dried on paper and stored for later analysis. The samples were evaluated using a JEM-1400Plus transmission electron microscope (JEOL, Tokyo, JP) [46,96].

To extract proteins from EVs from T. gondii or cells, EVs were initially obtained from the final step of ultracentrifugation of the previously described procedure. The enriched pellet was then resuspended in 20 µL of lysis buffer containing 0.25 M Tris HCl (pH 6.8), 40% glycerol, 8% sodium dodecyl sulfate (SDS), and 5% 2-mercaptoethanol, and incubated at 95 °C for 5 min. For protein quantification, 1 µL of each sample was mixed with a fluorescent reagent and measured using a Qubit^®^ 4.0 fluorometer (Thermo Fisher Scientific, Waltham, USA) at a wavelength range of 485 to 590 nm. After quantification, 0.04% bromophenol blue was added to the samples.

Approximately 10 μg of proteins were applied to a 12% polyacrylamide gel (8.6 × 6.7 cm), and the volume was adjusted with sample buffer. Electrophoresis was performed at 10 V/cm for 40 min until the proteins migrated 2 cm into the gel. The gel was stained with silver nitrate for visualization and then destained with sodium thiosulfate (100 mM) and potassium ferricyanide (30 mM), followed by incubation with ABC solution (50 mM). After dehydration with absolute ethanol, the samples were concentrated in a SpeedVac. They were then treated with a reduction buffer (10 mM DTT) at 60 °C, and, after the supernatant was removed, with an alkylation buffer (55 mM iodoacetamide) for 45 min, protected from light. The samples underwent enzymatic digestion using a digestion buffer (50 mM ABC) and trypsin (12.5 ng/μL), and were incubated for 18 h at 37 °C. After digestion, the peptides were extracted using a solution of water, 3% TFA and 30% acetonitrile [46].

Peptides were purified using Stage Tips, equilibrated with methanol (LC-MS grade) and solution A (water with 0.1% formic acid). The sample was applied and centrifuged at 1000× *g* for 5 min. Peptides were eluted with solution E (0.1% formic acid and 40% acetonitrile). The eluted peptides were concentrated in a SpeedVac and diluted in AD solution (0.1% formic acid, 5% DMSO, 5% acetonitrile). The analysis was performed on a liquid chromatography system coupled to an LTQ-Orbitrap XL ETD mass spectrometer (Thermo Scientific, USA). Data were processed using MaxQuant v2.6.5.0 and Perseus 1.6.10.43 softwares [46].

### 4.4. Extracellular Vesicle Cytotoxicity Assay

HBMEC cells were seeded into 96-well plates at a density of 4.0 × 10^3^ cells per well (~50% confluence). Cells were maintained for 24 h with 100 μL of DMEM per well, at 37 °C in a 5% CO_2_ atmosphere. After this period, the culture medium was replaced with a new medium containing half the original concentration of FBS, and EVs from tachyzoites were added at concentrations of 5, 10, 20, and 40 μg/mL for 24 h. After incubation, the medium was removed, and 100 μL of MTT solution (3-[4,5-dimethylthiazol-2-yl]-2,5-diphenyl tetrazolium bromide) at a concentration of 5 μg/mL was added to each well. The plate was then incubated for another four hours to protect it from light. The MTT solution was removed and 100 μL of isopropanol was added to each well. The plate was placed on a shaker for five minutes to homogenize the solution. The absorbance was measured at 550 nm using a Synergy H1 Hybrid Reader (BioTek Instruments, Inc., Winooski, VT, USA) [101].

### 4.5. Assessment of Gene Expression in Endothelial Cells Infected by T. gondii

HBMEC and HUVEC cells were seeded into 24-well plates at a density of 2.0 × 10^4^ cells per well (~50% confluence) and incubated in DMEM and EBM™-2 culture medium, respectively, at 37 °C in a 5% CO₂ atmosphere for 24 h. Tachyzoites maintained in NHDF cells were inoculated into HBMEC and HUVEC cells at MOI of 5. Plates were incubated for periods of 1, 2, 4, 8, 16, and 24 h. After each incubation period, RNA was extracted from the cells using the RNeasy^®^ Kit (Qiagen, Hilden, Germany) and stored at −80 °C. RNA quantification and integrity were assessed using a Qubit^®^ 4.0 fluorometer (Thermo Fisher Scientific, Waltham, MA, USA) with a fluorescent reagent, measured at 630–680 nm. Integrity was evaluated using a NanoDrop™ One spectrophotometer (Thermo Fisher Scientific, Waltham, MA, USA). The A260/A280 and A260/230 ratios were used to determine purity, with acceptable ranges of ~2.0 and 2.0–2.2, respectively. Approximately 100 ng of total RNA was reverse transcribed into cDNA using SuperScript™ II Reverse Transcriptase (Thermo Fisher, Waltham, MA, USA) following the manufacturer’s instructions. The final product was stored at 4 °C for subsequent use [24,25,52].

### 4.6. Quantification of Transcripts by qPCR

The qPCR reaction included sense and antisense primers (300 nM each), 5 μL of 2× SYBR™ Select Master Mix (Applied Biosystems, USA), 1 ng of cDNA, and ultrapure water to make up a final volume of 10 μL. Reactions were carried out in 96-well plates (Roche Life Science, Switzerland) and analyzed using a LightCycler^®^ 96 qPCR system (Roche Life Science, Basel, Switzerland). The qPCR cycle settings followed the SYBR™ Select Master Mix manufacturer’s protocol. All genes were analyzed using 50 cycles, and primers were designed for an optimal annealing temperature of 60 °C. Normalization was based on *ACTINB* and *GAPDH* standard curves (R^2^ ≈ 1.0), and quantification was relative to this control using the 2^−ΔΔCt^ method. *ACTINB* was chosen as the reference gene. Each gene was analyzed in three biological replicates, with each biological replicate having three technical replicates [52,102]. The sequence of each primer is presented in the Appendix A.

### 4.7. Infection Assay

HBMEC and HUVEC cells were seeded into 24-well plates with sterile coverslips placed at the bottom of the wells at a density of 2.0 × 10^4^ cells per well (~50% confluence). Cells were maintained for 24 h with 1 mL of DMEM (HBMEC) or EBM™-2 (HUVEC) medium per well at 37 °C in a 5% CO_2_ atmosphere. Cells were then treated with HB-EV, HBTg-EV, HBTg-SEC, or Tg-EV at concentrations corresponding to vesicles secreted by 1.0 × 10⁵ tachyzoites (~7.4 ng). TNF-α (10 ng/mL) was used as a positive control, and PBS as a negative control. After an hour, cells were washed with PBS, fixed with 1:1 methanol/acetone for 15 min, and blocked with 1% PBS/BSA (AMRESCO, Solon, OH, USA) for four hours. Cells were then incubated with a specific antibody (IgG) for *T. gondii* anti-Surface Antigen 1 (SAG1) (1:200) produced in mice and kindly provided by Tiago Mineo (Universidade Federal de Uberlândia) for one hour, followed by a secondary anti-mouse IgG (1:600) conjugated to Alexa Fluor^®^ 488 for 45 min. For identification of nuclei and cytoplasm, DAPI and T-1824 (Evans Blue) were added at final concentrations of 10 μM and 0.1%, respectively. Coverslips were mounted on slides using propyl 3,4,5-trihydroxybenzoate and sealed with varnish. Images were captured using an inverted deconvolution microscope (Leica DMI6000, Leica Geosystem, Heerbrugg, CH) [103,104].

### 4.8. Assessment of T. gondii Infection and Adhesion Molecule Expression

HBMEC and HUVEC cells were seeded into T25 flasks at a density of 5.0 × 10^5^ cells per flask and incubated in DMEM and EBM™-2, respectively, at 37 °C and 5% CO₂ for 24 h. Cells were infected with 2.0 × 10^6^ *T. gondii* tachyzoites (MOI-5) for 24 h. TNF-α (10 ng/mL) was used as a positive control, and PBS as a negative control. Cells were detached using EDTA (1.5 mM) and a cell scraper, centrifuged at 300× *g* for 10 min, and resuspended in PBS. Primary antibodies anti-SAG1 (1:200), anti-Intercellular Adhesion Molecule (ICAM-1) (1:1000), and anti-Vascular Adhesion Molecule (VCAM-1) conjugated to APC (1:1000) were incubated for one hour. Secondary antibodies, anti-mouse IgG conjugated to Alexa Fluor^®^ 488 and Alexa Fluor^®^ 633, were used for ICAM-1 and SAG1 labeling. Cells were analyzed using a FACSCanto II flow cytometer (Becton Dickinson, Franklin Lakes, NJ, USA), and data were processed with FlowJo v10.6.1 software (Becton Dickinson, Franklin Lakes, NJ, USA) [105].

### 4.9. Assessment of T. gondii by Electron Microscopy

For Transmission Electron Microscopy (TEM), tachyzoites cultured in NHDF were collected, washed twice in 10 mL of PBS, and suspended in a fixation solution composed of 3% glutaraldehyde diluted in 0.1 M sodium cacodylate buffer, pH 7.0. The samples were then washed in several cycles of buffer changes and fixed in 1% osmium tetroxide for one hour. Afterward, the samples were washed again in buffer solution and transferred to permeable baskets. Dehydration was performed by sequential immersion in 50%, 70%, 80%, 95%, and 100% ethanol solutions. Following dehydration, the samples were embedded in a 1:1 resin/acetone solution, which was changed after eight hours and incubated at 60 °C for two days. After this period, ultrathin sections were prepared and stained with uranyl acetate and lead citrate [106].

For Scanning Electron Microscopy (SEM), tachyzoites cultured in NHDF were collected, washed twice in 10 mL of PBS, and suspended in a fixation solution composed of 3% glutaraldehyde diluted in 0.1 M sodium cacodylate buffer, pH 7.0. The samples were then washed in several cycles of buffer changes and fixed in 1% osmium tetroxide for one hour. After another wash in buffer solution, the samples were transferred to permeable baskets. Dehydration was performed by immersion in 50%, 70%, 80%, 95%, and 100% ethanol solutions. After the final immersion, the ethanol was replaced by liquid CO_2_. Metallization of the samples was carried out by spraying gold particles [42].

### 4.10. Statistical Analysis

The statistical evaluation of the results was carried out based on the data obtained from each experiment. Normality and homoscedasticity of the samples were assessed using the Shapiro–Wilk and Levene tests, respectively. The *t*-test was used to compare means between two unpaired parametric variables. For multiple comparisons of unpaired samples with a normal Gaussian distribution (parametric), analysis of variance (ANOVA) followed by Tukey’s test was applied. For non-parametric samples, multiple comparisons were performed using the Kruskal–Wallis test followed by Dunn’s test. Heatmaps were constructed by applying metric space between measurements, using Euclidean distance. Sample groups were based on the Pearson correlation. The breakdown of the selected statistical tests can be found in the figure legends. All statistical analyses were performed using GraphPad Prism 8.0 (GraphPad Software, Inc., Boston, MA, USA) and RStudio 4.2.2 (RStudio Inc., Boston, MA, USA). Analyses with a *p*-value (probability of significance) equal to or lower than 0.05 were considered statistically significant, with the following indicators: * for *p* < 0.05; ** for *p* < 0.01; *** for *p* < 0.001.

## 5. Conclusions

These findings highlight the distinct responses of HBMEC and HUVEC to *T. gondii* infection, with HUVEC exhibiting stronger pro-inflammatory and transcriptional changes, while HBMEC showed more subdued responses. The role of *T. gondii* EVs in modulating endothelial cell viability and gene expression underscores the complexity of host–parasite interactions, paving the way for further investigations into the molecular mechanisms involved.

## Figures and Tables

**Figure 1 ijms-26-02640-f001:**
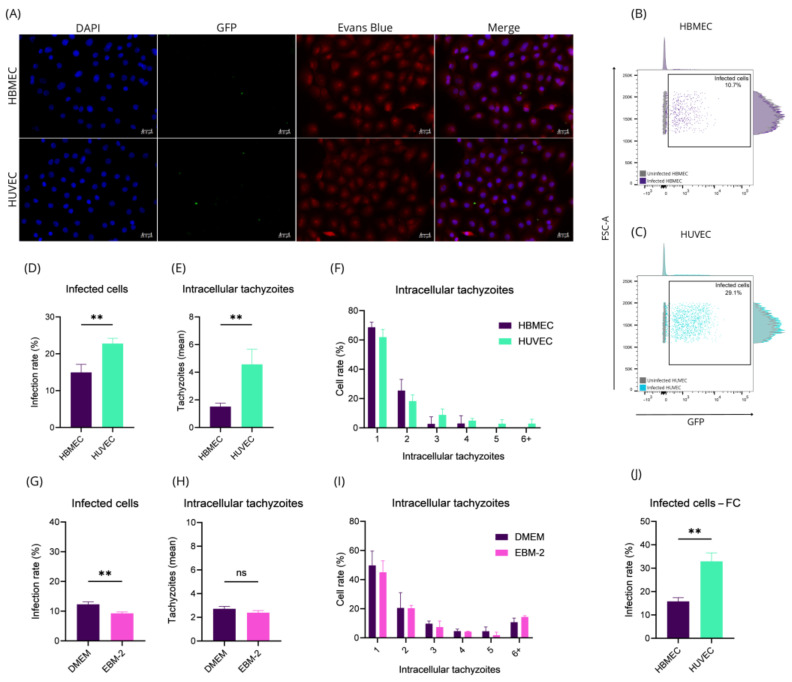
Infection rate of HBMEC and HUVEC is different. HBMEC and HUVEC were infected with tachyzoites at MOI-5 for 1 h. After that, the cells were washed, fixed, labeled with DAPI, a primary antibody specific to SAG1, and a secondary antibody specific to mouse IgG conjugated to Alexa Fluor^®^ 488. Infection rate was calculated by counting 5 fields per biological replicate using a 40× objective (**A**) and expression as percentage of infected cells (**D**,**G**), intracellular tachyzoite average (**E**,**H**), and distribution of the number of intracellular tachyzoites (**F**,**I**). Another group of infected or uninfected HBMEC and HUVEC was evaluated by flow cytometry (FC) (**B**,**C**) with the results being expressed graphically (**J**). Graphs expressed as mean ± standard error. Welch’s *t*-test (**D**), *t*-test (**E**,**G**,**H**,**J**), multiple *t*-test with Holme–Sidak correction (**F,I**). **: *p*-value < 0.01; ns: not significant.

**Figure 2 ijms-26-02640-f002:**
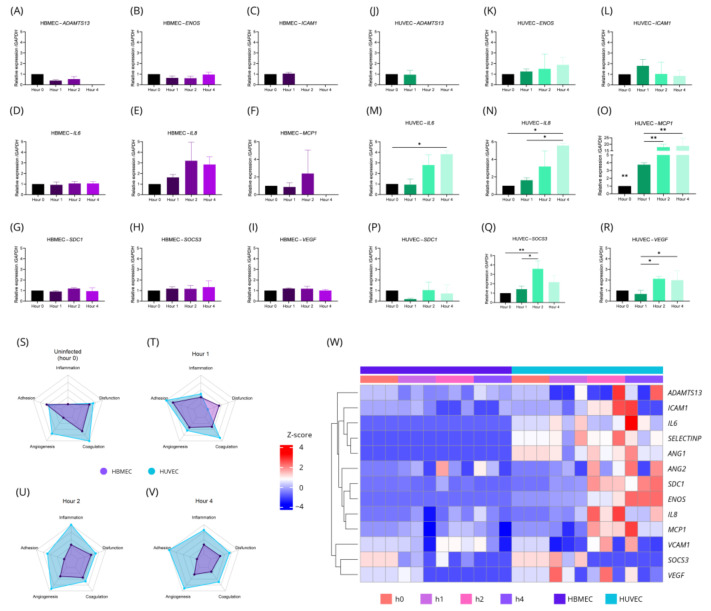
Transcriptional changes in HBMEC and HUVEC in infection with *T. gondii* are distinct. HBMEC and HUVEC were infected by *T. gondii* and left for 1 (h1), 2 (h2), or 4 h (h4). Next, the RNA from the cells was extracted and converted to cDNA and the transcription pattern was evaluated by qPCR and presented in the form of bar chart (**A**–**R**), radar chart (**S**–**V**), or a heatmap (**W**). Analysis of uninfected cells was considered as time 0 (h0). Clustering in the heatmap was performed by Euclidean distancing after normalization by z-score. Data were obtained from three biological and technical replicates and are expressed as the mean ± standard deviation. ANOVA/Tukey (**A**–**R**). *: *p*-value < 0.05; **: *p*-value < 0.01.

**Figure 3 ijms-26-02640-f003:**
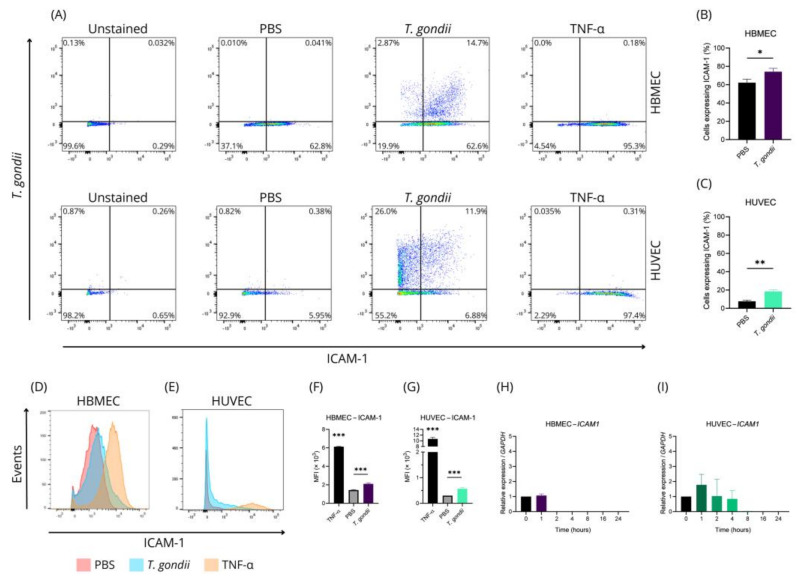
*T. gondii* infection stimulates ICAM-1 expression in endothelial cells. HBMECs and HUVECs were incubated with TNF (10 ng/mL), PBS, or *T. gondii* (MOI of 5) for 24 h. Following this, cells were stained with antibodies specific for ICAM-1 and *T. gondii* and analyzed by flow cytometry (**A**). Quantification of ICAM-1 expression in cells treated with PBS and *T. gondii* is shown in bar graphs (**B**,**C**). The mean fluorescence intensity (MFI), based on the geometric mean of signal intensity, is displayed in a histogram (**D**,**E**) and summarized in bar graphs (**F**,**G**). ICAM-1 expression in HBMECs and HUVECs was also assessed at 1, 2, 4, 8, 16, and 24 h post-infection (**H**,**I**). Data are presented as mean ± standard deviation from three biological and technical replicates. *t*-test (**B**,**C**); ANOVA/Tukey (**F**–**I**). *: *p*-value < 0.05; **: *p*-value < 0.01; ***: *p*-value < 0.001.

**Figure 4 ijms-26-02640-f004:**
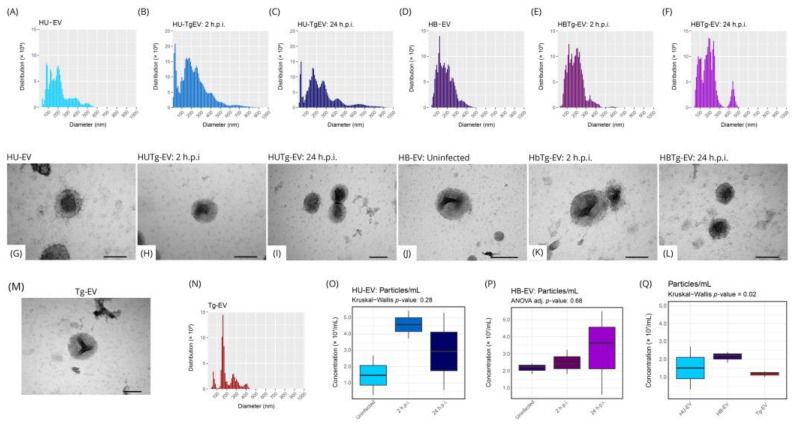
Minimal differences in EVs from HUVEC and HBMEC after infection with *T. gondii*. EVs were isolated from both infected and uninfected HUVEC and HBMEC cells following *T. gondii* exposure, using differential centrifugation. EV distribution was characterized by NTA (**A**–**F**,**N**) and morphology assessed by TEM (**G**–**L**,**M**). EV concentration comparisons before and after infection with *T. gondii* (**O**–**Q**). Statistical analyses were performed using Kruskal–Wallis/Dunn and ANOVA/Tukey tests (**O**–**Q**). Data are based on three biological replicates and are presented as mean (**A**–**F**,**N**) and mean ± standard deviation (**O**–**Q**). Scale bars in size 100 nm (**G**–**M**).

**Figure 5 ijms-26-02640-f005:**
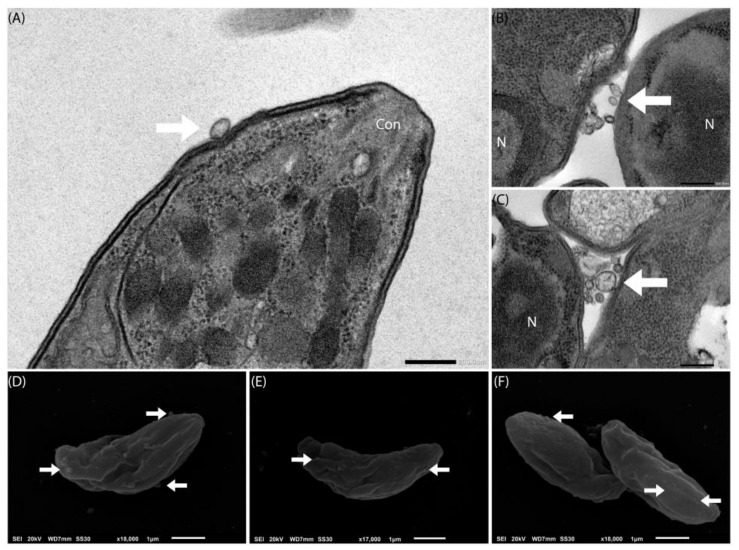
*T. gondii* produces particles compatible with EVs. Free parasites in the 1.0 × 10^8^ population were obtained from cultivation in NHDF. Fixation was performed with glutaric aldehyde (3%) and negative counterstaining with uranyl and lead acetate. The tachyzoites were sequentially observed under a transmission (**A**–**C**) and scanning (**D**–**F**) electron microscope. Arrows indicate particles indicative of being EVs. Scale bars in size 200 nm (**A**–**C**) and 1 μm (**D**–**F**).

**Figure 6 ijms-26-02640-f006:**
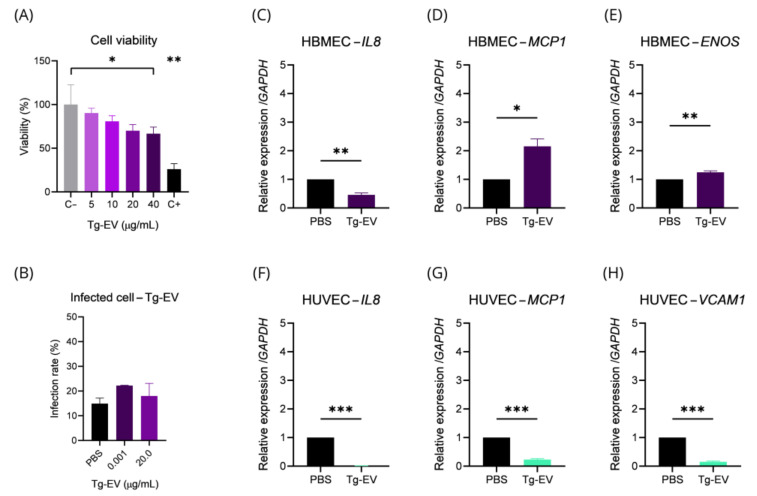
Tg-EV modulates gene expression in endothelial cells. Cell viability of HBMECs cultured in 96-well plates and incubated with 5, 10, 20, or 40 μg of Tg-EV was assessed using the MTT assay (**A**). Subsequently, infection assay was conducted by infecting cells with *T. gondii* at MOI of 5, followed by incubation with 1.0 ng/mL or 20 μg/mL of Tg-EV, with quantification by Fluorescence Microscopy (**B**). Additionally, we evaluated the gene expression of IL8, MCP1, ENOS, and VCAM1 in HBMECs (**C**–**E**) and HUVECs (**F**–**H**) incubated with Tg-EV (1.17 ng/mL) for 24 h. Data are presented as mean ± standard deviation from three biological and technical replicates. ANOVA/Tukey (**A**,**B**), *t*-test (**C**–**H**). *: *p*-value < 0.05; **: *p*-value < 0.01; ***: *p*-value < 0.001.

**Figure 7 ijms-26-02640-f007:**
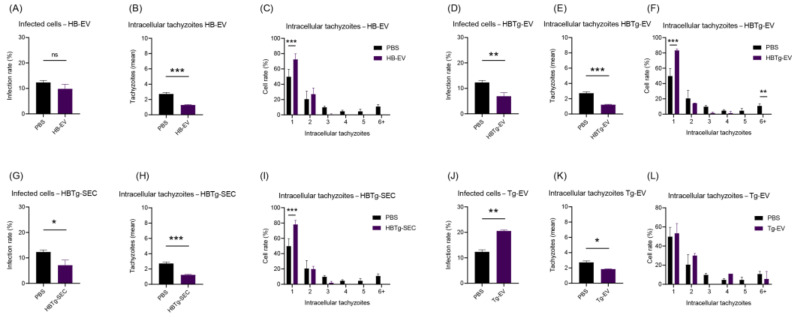
EVs secreted by infected or uninfected HBMECs promote a protective effect on HBMEC infection by *T. gondii*. About 6.4×10^3^ HBMECs were cultured in 96-well plates with 100 μL DMEM. Then, they were incubated for 24 h with 1.17 ng HB-EV (**A**–**C**), HBTg-EV (**D**–**F**), HBTg-SEC (**G**–**I**) or Tg-EV (**J**–**L**). Cells were infected (MOI-5) with *T. gondii* ME-49, washed after 1 h, fixed, stained (DAPI, Evans Blue), and analyzed by Fluorescence Microscopy. All cells in three different fields per experimental unit were counted. Data obtained from three biological replicates and expressed as the mean ± standard deviation. Simple *t*-test (**A**,**B**,**D**,**E**,**G**,**H**,**J**,**K**) and multiple (**C**,**F**,**I**,**L**); *: *p*-value < 0.05; **: *p*-value < 0.01; ***: *p*-value < 0.005; ns: not significant.

**Figure 8 ijms-26-02640-f008:**
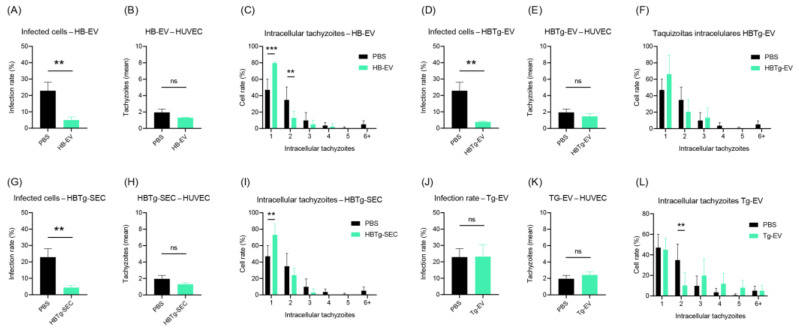
EVs secreted by HBMECs promote a protective effect on HUVEC infection by *T. gondii*. About 6.4 × 10^3^ HUVECs were cultured in a 96-well plate with 100 μL of EBM-2. Then, they were incubated for 24 h with 1.17 ng HB-EV (**A*–*C**), HBTg-EV (**D**–**F**), HBTg-SEC (**G**–**I**) or Tg-EV (**J**–**L**). Cells were infected (MOI-5) with T. gondii ME-49, washed after 1h, fixed, stained (DAPI, Evans Blue), and analyzed by Fluorescence Microscopy. All cells in three different fields per experimental unit were counted. Data obtained from three biological replicates and expressed as the mean ± standard deviation. Simple *t*-test (**A**,**B**,**D**,**E**,**G**,**H**,**J**,**K**) and multiple (**C**,**F**,**I**,**L**); **: *p*-value < 0.01; ***: *p*-value < 0.005; ns: not significant.

**Figure 9 ijms-26-02640-f009:**
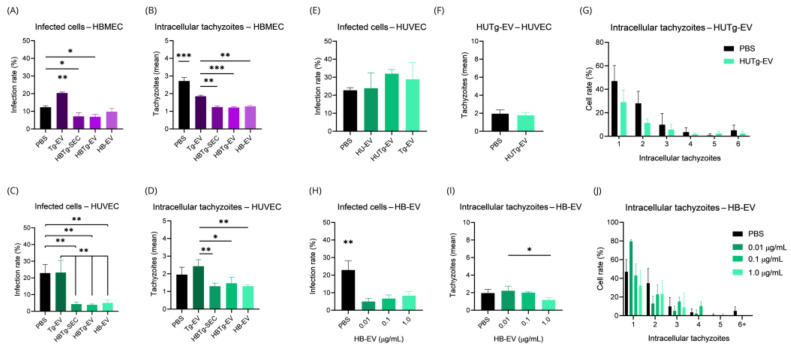
Infection rate of HBMEC and HUVEC is affected by different EVs and HB-EV concentrations. About 6.4 × 10^3^ HBMEC and HUVEC were cultured in a 96-well plate with 100 μL of DMEM or EBM™-2. They were then incubated for 24 h with 1.17 ng/mL of Tg-EV, HBTg-SEC, HBTg-EV, or HB-EV (**A**–**D**). HUVEC were also incubated with HU-EV (**E**), Tg-EV (**E**), HUTg-EV (**E**–**G**), or 0.01, 0.1, or 1.0 μg/mL of HB-EV (**H**–**J**). Cells were infected (*T. gondii*, MOI-5), washed after 1 h, fixed, stained (DAPI, Evans Blue), and analyzed for infection rate and tachyzoite count by Fluorescence Microscopy. All cells in three different fields per experimental unit were counted. Data obtained from three biological replicates are expressed as the mean ± standard deviation. Simple *t*-test (**F**); Multiple *t*-test (**G**,**J**). ANOVA/Tukey (**A**–**D**,**E**,**H**,**I**). *: *p*-value < 0.05; **: *p*-value < 0.01; ***: *p*-value < 0.005.

**Figure 10 ijms-26-02640-f010:**
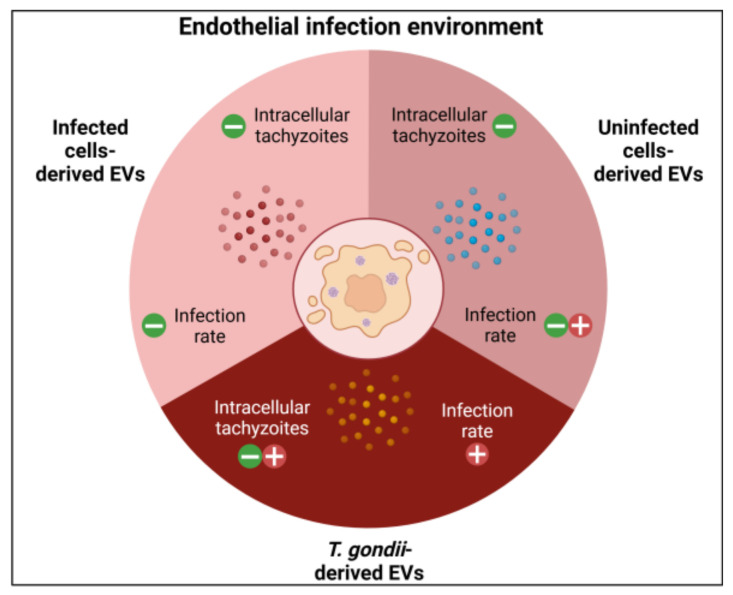
Graphical summary. The different types of EVs in the infectious context of HBMEC by *T. gondii* and their effects on the infection rate of HBMEC and HUVEC. The symbol ‘–’ represents reduction, while ‘+’ indicates increase. Created in BioRender. Albrecht, L. (2025) https://BioRender.com/n23l151, acessed on 25 February 2025.

**Figure 11 ijms-26-02640-f011:**
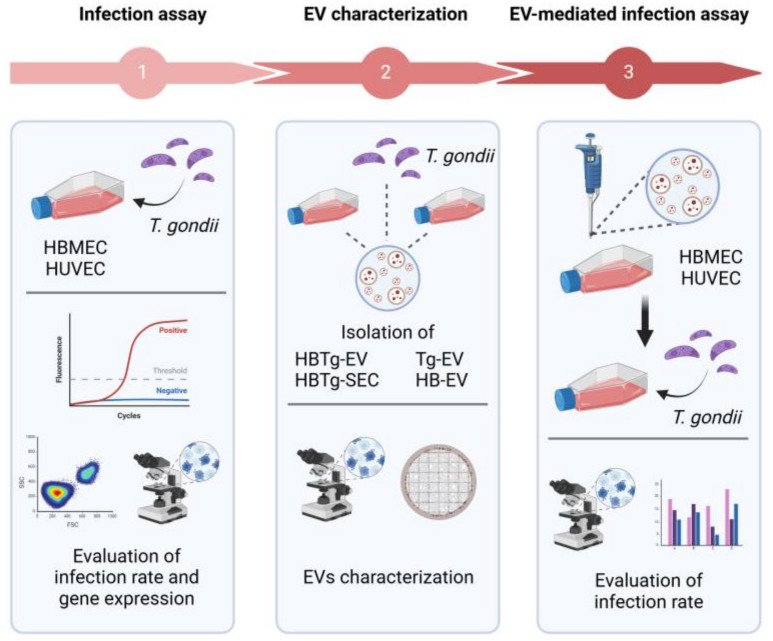
Experimental diagram. HBMEC and HUVEC were infected by *T. gondii* at MOI-5 and the infection rate was assessed by Fluorescence Microscopy and Flow Cytometry (1). EVs from HBMEC infected (HBTg-EV) or not infected (HB-EV), from *T. gondii* (Tg-EV) and secretome of infected HBMEC (HBTg-SEC) were obtained by differential centrifugation. The particles obtained were evaluated by TEM, SEM, and NTA (2). Next, the infection assessment of HBMEC and HUVEC previously incubated with EVs or secretome was carried out (3). Created in BioRender. Albrecht, L. (2025) https://BioRender.com/o87d712. Acessed on 25 February 2025.

## Data Availability

All data generated or analyzed during this study are included in this published article and its Appendix A.

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
