# Peer review of "Human Brain Endothelial Cell-Derived Extracellular Vesicles Reduce Toxoplasma gondii Infection In Vitro in Human Brain and Umbilical Cord Vein Endothelial Cells"

_ijms, 2025, doi:10.3390/ijms26062640_

Round 1
Reviewer 1 Report
Comments and Suggestions for Authors
In general, the work is well written and understandable. But there are points that need to be resolved:
On line 65-66, explain the origin of EVs from parasite excretion/secretion.
In the methods, on line 652, what is the origin of the HBMEC cell line? Is it a standard strain or one maintained in the laboratory? Explain more about the origin of the cells.
Reviewer 2 Report
Comments and Suggestions for Authors
Dear Editorial Board,
In the manuscript titled " Brain endothelial cell-derived extracellular vesicles reduce toxoplasma gondii infection in brain and umbilical cord vein endothelial cells" the authors investigate the interaction between Toxoplasma gondii and endothelial cells, highlighting distinct responses in human brain microvascular endothelial cells (HBMEC) and human umbilical vein endothelial cells (HUVEC). While the data presented are intriguing, particularly in demonstrating how extracellular vesicles (EVs) can modulate infection rates, I recommend conducting additional analyses before considering the manuscript for publication. The main limitation of this study is the lack of an in vivo experiments that could boost the data obtained. Furthermore, the manuscript is suffering from many problems including text statements that do not reflect the data shown and so forth. I found this paper difficult to follow and understand. This pertains mostly to the first half of the paper and the discussion.
The title is unclear compared to the content of the paper. The authors should specify in the title that this is an in vitro study.
Methods:
- Line 728: The authors describe the protein extraction method, but no reference or detailed methodology is provided. The authors should include a description of the protein extraction protocol.
- Line 834: Although the meaning of the asterisks after statistical analysis is reported in the figure legends, this should also be clarified in this section of the text.
- The authors use Evans Blue in several figures, but they do not explain its purpose or whether it indicates compromised cell integrity following infection. Similarly, the use of the surface antigen (SAG1) antibody is not explained in detail; the full name and its relevance (as a specific antibody for Toxoplasma gondii) should be stated.
Figure 1:
- Panel A: The GFP staining is challenging to visualize. The authors should replace these images and include insets with higher magnification to clearly show Toxoplasma gondii presence in the cells.
- Panels B and C: The text in the graphs is illegible. The authors should replace these graphs with improved formatting.
- Line 109: There is an incomplete sentence. The authors should revise and verify the text.
Figure 2:
- The presentation of results in histograms is difficult to follow. A table summarizing the data might be more helpful for the reader.
- The figure legend does not specify the statistical analyses performed or the significance of the asterisks. This should be clarified in both the legend and the methods section.
Line 206: The formatting of T. gondii should be consistent throughout the text (e.g., Lines 226, 231).
Figure 4 (Panels G-L):
- It is unclear what these images are intended to represent. The authors state they are used to evaluate cell morphology, but at this magnification, no differences can be observed, and no quantitative analysis has been performed. These images might be better suited as supplementary data to avoid reader confusion.
Line 210: The authors should indicate the average sizes of extracellular vesicles (EVs) obtained under different conditions.
Line 219: The authors state, “The EV distribution is also presented as a conditioned histogram (M),” but this does not seem to match what is shown in the figure.
Line 227: The text mentions that panel M contains an electron microscopy image, but it appears to have the same resolution as panels in Figure 4 (G-L). The authors should clarify this.
Line 253: The authors claim, “…differences in EV size, concentration, and protein markers underscore…,” but no corresponding analyses have been performed. The authors should evaluate these parameters. Are the vesicle sizes obtained through electron microscopy comparable with previous analyses?
Figure 6: The histogram in panel C+ should be clearly labeled in the text and figure legend.
Section 2.4:
- The authors describe using HBMEC as a cellular model but also report using HUVEC. However, viability and infection percentages are based solely on HBMEC data. The authors should evaluate the same effects (shown in Figure 6 A and B) in HUVEC as well.
Line 307-309: This is a description of a method and should be moved to the methods section for clarity.
Lines 316-335: The p values are in a larger font than the rest of the text. This should be standardized throughout the manuscript.
Discussion:
- The discussion is overly long and difficult to follow. Much of the initial section reads more like an introduction rather than a discussion of the results. Similar issues are present in other parts of the text. The authors should streamline and significantly condense the discussion to improve readability and clarity.
Round 2
Reviewer 2 Report
Comments and Suggestions for Authors
The authors have adequately addressed my concerns, and I have no further comments.